# Rare-Metal Pegmatite Deposits of the Kalba Region, Eastern Kazakhstan: Age, Composition and Petrogenetic Implications

**Sergey V. Khromykh** [1,2,*], **Tatiana A. Oitseva** [3], **Pavel D. Kotler** [1,2], **Boris A. D'yachkov** [3], **Sergey Z. Smirnov** [1,4], **Alexey V. Travin** [1], **Alexander G. Vladimirov** [1], **Ekaterina N. Sokolova** [1,2], **Oxana N. Kuzmina** [3], **Marina A. Mizernaya** [3] and **Bakytgul' B. Agaliyeva** [3]

[1] V.S. Sobolev Institute of Geology and Mineralogy Siberian Branch of the Russian Academy of Sciences, 630090 Novosibirsk, Russia; pkotler@yandex.ru (P.D.K.); ssmr@igm.nsc.ru (S.Z.S.); travin@igm.nsc.ru (A.V.T.); vladimir@igm.nsc.ru (A.G.V.); e.post@ngs.ru (E.N.S.)

[2] Department of Geology and Geophysics, Novosibirsk State University, 630090 Novosibirsk, Russia

[3] D. Serikbayev East Kazakhstan State Technical University, Ust-Kamenogorsk 070000, Kazakhstan; tatiana.oitseva@gmail.com (T.A.O.); bdyachkov@mail.ru (B.A.D.); kik_kuzmins@mail.ru (O.N.K.); mizernaya58@bk.ru (M.A.M.); agalieva_00@mail.ru (B.B.A.)

[4] Faculty of Geology and Geography, Tomsk State University, 634050 Tomsk, Russia

\* Correspondence: serkhrom@igm.nsc.ru; Tel.: +7-9-139-093-079

**Abstract:** The paper presents new geological, mineralogical, and isotope geochronological data for rare-metal pegmatites in the Kalba granitic batholith (Eastern Kazakhstan). Mineralization is especially abundant in the Central-Kalba ore district, where pegmatite bodies occur at the top of large granite plutons and at intersections of deep faults. The pegmatites contain several successive mineral assemblages from barren quartz-microcline and quartz-microcline-albite to Li-Cs-Ta-Nb-Be-Sn-bearing cleavelandite-lepidolite-spodumene. Ar-Ar muscovite and lepidolite ages bracket the metallogenic event between 291 and 286 Ma. The pegmatite mineral deposits formed synchronously with the emplacement of the phase 1 Kalba granites during the evolution of hydrous silicate rare-metal magmas that are produced by the differentiation of granite magma at large sources with possible inputs of F and rare metals with fluids.

**Keywords:** rare-metal pegmatites; Ar-Ar isotope age; Kalba granite batholith; Eastern Kazakhstan

## 1. Introduction

Deposits of rare metals (Li, Rb, Cs, Be, Zr, Hf, Nb, Ta, Sn, Mo, and W) that are genetically and spatially related to magmatism are widespread in the Central Asian orogenic belt. The metals (Li, Cs, Ta, Nb, Be, Sn) most often occur in pegmatites that are derived from granitoids of different compositions, in Asia as well as elsewhere worldwide [1–10]. Metal enrichments in the parent magma may be high or within average values for the upper continental crust. Metal-bearing pegmatites often form relatively independent fields of veins or dikes, which are notably younger (up to hundreds of million years) than the enclosing granites [3,6,11]. The evolution of granitic magmas leading to rare-metal mineralization may be driven by different mechanisms, including effects from mantle sources, as in the case of Li-Cs-Ta pegmatites and coexisting Li-F granites in Central Asia [5,12,13].

The compositions and ages of coexisting rare-metal pegmatites and granites have important petrogenetic implications for the formation mechanisms of mineralization. This paper presents the recent mineralogical and geochronological data for the Kalba rare-metal pegmatites in Eastern Kazakhstan.

## 2. Geological Background

### 2.1. Tectonic Position

The territory of Eastern Kazakhstan belongs to the central Late Paleozoic Altai orogen that results from a collision between Siberia and Kazakhstan, which were separated by the Ob'-Zaisan paleo-ocean ([14–16] and etc.). The oceanic basin accommodated Late Devonian-Early Carboniferous clastic deposits that were produced by erosion of volcanic rocks on the active continental margin (Rudny Altai terrane). The continents collided and the ocean closed in the latest Early Carboniferous, whereby the clastic sediments became stacked to a total thickness of 10–12 km and metamorphosed. The area has been interpreted as the Kalba-Narym turbidite terrane, a remnant of a continental margin basin upon oceanic crust [15,17,18]. The terrane is bordered by the Terekta fault in the southwest, at the boundary with the Char terrane, and by the Irtysh shear zone in the northeast, next to the Rudny Altai terrane (Figure 1).

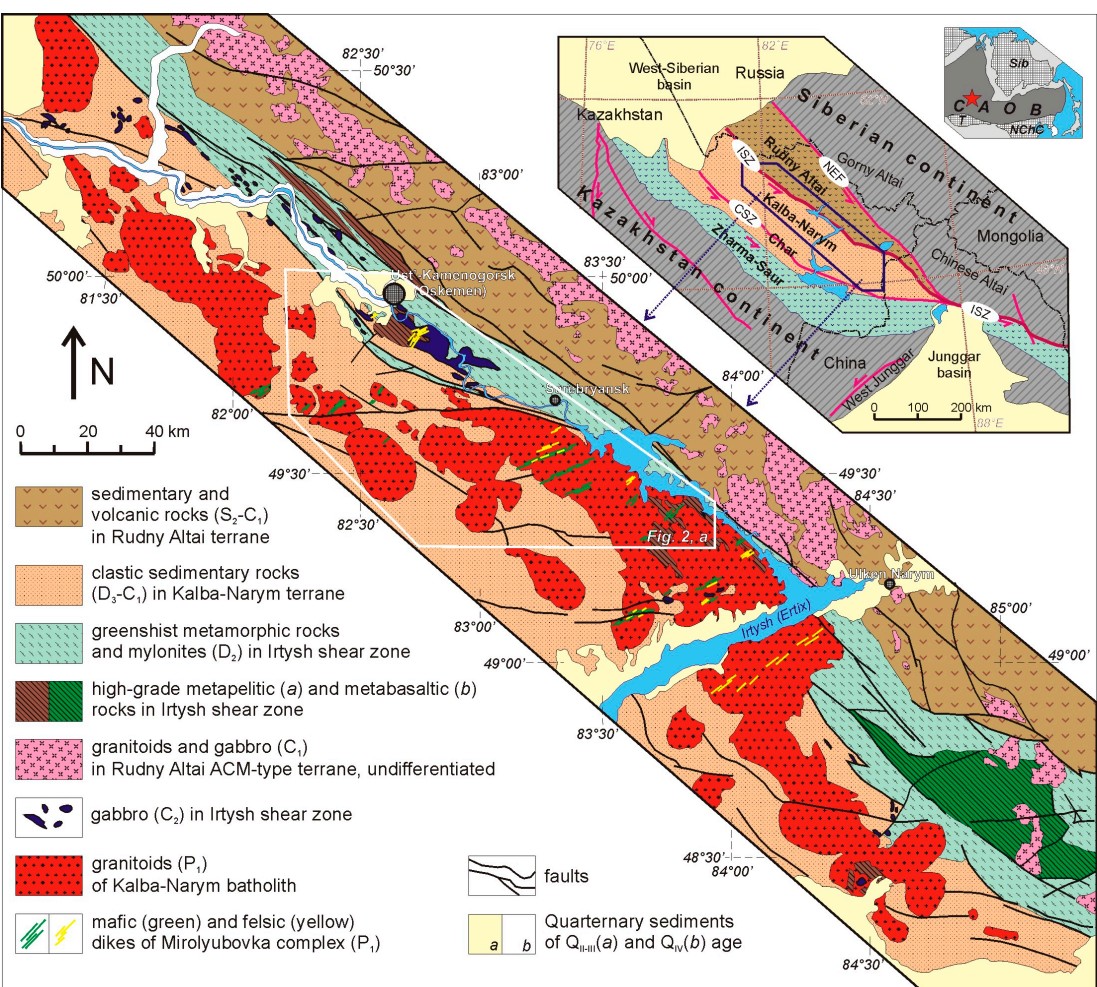

**Figure 1.** Simplified geology of the Kalba-Narym terrane and the Kalba batholith. Inset shows the location of main terranes in the collisional system. Main faults: Char shear zone (CSZ), Irtysh Shear Zone (ISZ), and North-East Fault (NEF).

The Irtysh shear zone originated in the Early Carboniferous in the beginning of accretionary-collisional processes and underwent major deformation events at 360, 320, 300, 280, and 260 Ma [19–21]. Most of the rocks in the zone appear in the present erosion cutout as greenschist facies mylonites and blastomylonites, although some blocks are composed of high-grade metamorphic

rocks (Figure 1). The zone may be a deep metamorphosed part of the section of the Kalba-Narym terrane, according to isotope data [22].

## 2.2. Kalba Batholith

The Kalba batholith, the principal unit of the Kalba-Narym terrane, is an SE—NW 400 km long and 50 km wide belt that consists of several granitoid complexes. The Kalba granitoid magmatism has been explained by several models since the beginning of detailed studies in the 1960s–1970s ([23–25] and etc.). More constraints on its age and duration were obtained from recent geological and U-Pb geochronological data [18,26,27]. The oldest sporadic granodiorite and granite intrusions of the Kalguty complex and plagiogranite intrusions of the Kunush complex formed in the Late Carboniferous (308–303 Ma).

More than 70% of the batholith exposed in the present erosion cutout belongs to the Kalba complex, which was emplaced in two phases: (1) biotite granodiorite and granite at 297–288 Ma and (2) biotite and two-mica granite at 286–283 Ma. Phase 1 Kalba granites are outsized, mainly medium, and coarse grained. Their mineralogy consists of 30–35 vol.% plagioclase, 20–25 vol.% K-feldspar, 20–30 vol.% quartz, 10–15 vol.% biotite (the only mafic mineral), and typical granitic accessories of muscovite, apatite, ilmenite, garnet, zircon, fluorite, and tourmaline, as well as sporadic cassiterite, and molybdenite [23,24,28]. The major-oxide contents are 63 to 70 wt.% $SiO_2$, 6.4 to 8.9 wt.% total alkalis ($Na_2O + K_2O$), 1.1 to 2.7 wt.% CaO, 1.1 to 4.5 wt.% FeO, and 0.8 to 1.2 wt.% MgO; metals reach relatively high contents: 10–20 ppm Sn, 4 ppm Be, 250 ppm Li, and 1.7–2.5 ppm Ta, while the total of rare alkalis (Li + Rb + Cs) amounts to 480 ppm, which exceeds the average values in upper continental crust, UCC [29].

The less abundant phase 2 Kalba granites are equigranular fine or medium-grained rocks with similar percentages of K-feldspar and quartz (about 30 vol.% each), around 5 vol.% biotite and 3 vol.% muscovite; the accessory minerals are zircon, apatite, garnet, tourmaline, and fluorite. The major-oxide composition includes 70 to 74 wt.% $SiO_2$, 6.8 to 7.9 wt.% total alkalis ($Na_2O + K_2O$), 2 to 2.5 wt.% CaO, 1.1 to 2.8 wt.% FeO, and 0.3 to 0.8 wt.% MgO. The rare-metal contents are lower than in the phase 1 granites (4–6 ppm Sn and about average UCC quantities of Ta, Nb, and Be); the (Li + Rb + Cs) total is only 102 ppm. Mineralization mainly resides in Ta- and Be-quartz-albite-muscovite pegmatites (Kvartsevoye deposit), Sn-, Ta-, and Li-albite greisen (Karasu and Apogranitnoye occurrences), quartz-muscovite greisen, as well as quartz veins with cassiterite and wolframite of non-commercial value [30].

The 283–276 Ma Monastery complex comprises several large isometric intrusions of biotite and two-mica granite and leucogranite in the southwestern part of the Kalba batholith. They bear monazite mineralization and they are genetically related with barren pegmatites and W-bearing quartz veins [23,25,30].

The Kalba batholith mostly formed in the Early Permian [18,26] during the postorogenic evolution of the Altai collisional system in a setting of extension that was probably associated with mantle plume activity in the Tarim large igneous province [27].

## 2.3. Dike Belts

NE-trending belts of mafic and felsic dikes (Mirolyubovka complex) are the youngest igneous bodies in the Kalba-Narym terrane. They were previously timed as Late Permian-Triassic proceeding from a discordant geological position [23–25], but recent geochronological data indicate a synplutonic origin.

Mafic dikes (Figures 1 and 2a) consist of dolerite and lamprophyre with relatively high alkali contents (3.4 wt.% to 7.3 wt.% $Na_2O + K_2O$), 0.25 to 1.05 wt.% $P_2O_5$, and high concentrations of rare metals, fluorine, and boron: 3 to 7 ppm Li, 0.9 to 3.5 ppm Cs, 16 to 140 ppm Rb, 0.5 to 1 ppm Sn, 0.1 to 0.5 ppm Be, 5 to 18 ppm Nb, 0.5 to 1.3 ppm Ta, 120 to 140 ppm F, and 1 to 3 ppm B [23]. A mafic dike in the central part of the batholith has a U-Pb zircon age of 279 ± 3 Ma [27].

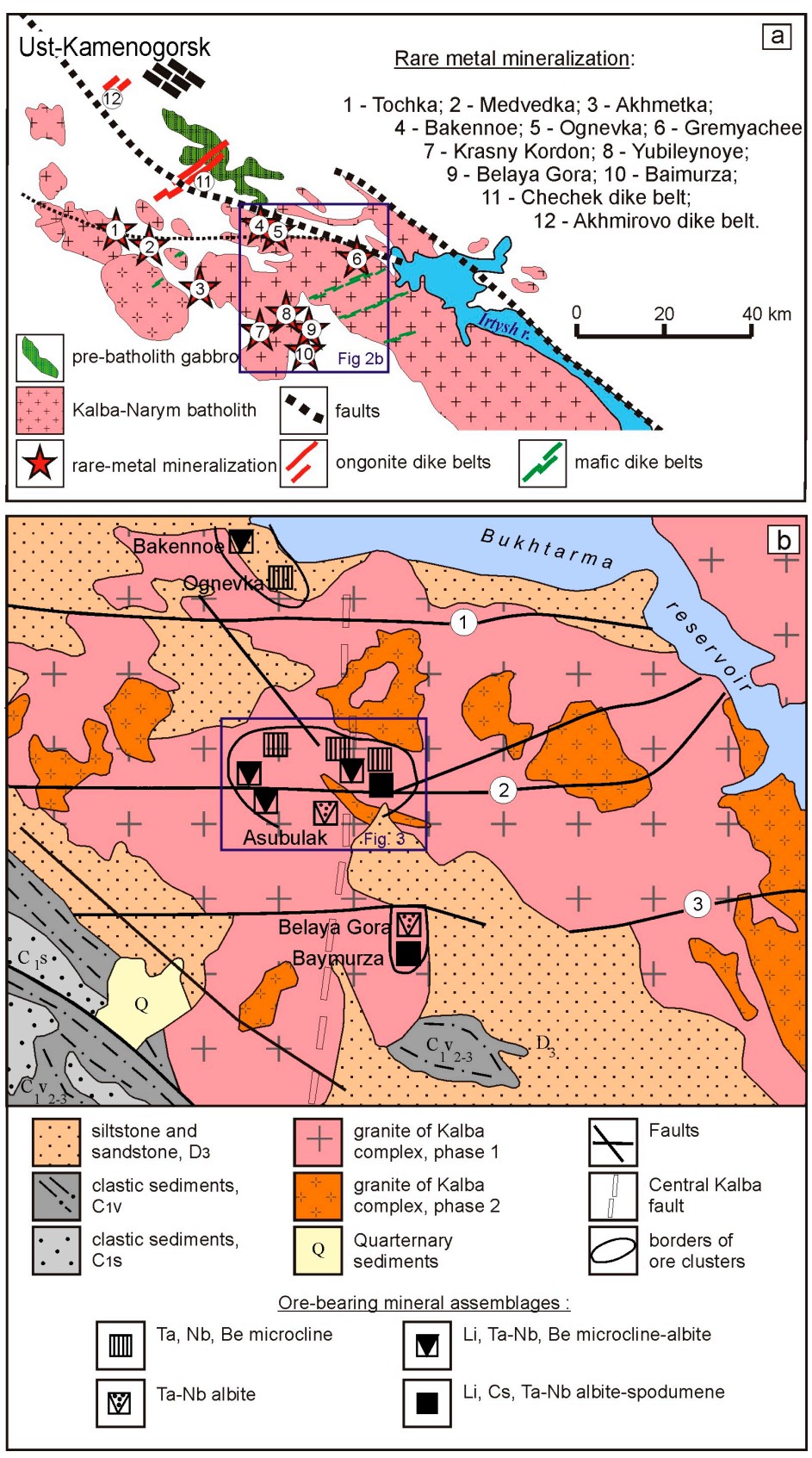

**Figure 2.** Local geology of rare-metal pegmatite deposits. (**a**) Location of rare-metal pegmatites and ongonite dikes in the central and northwestern Kalba batholith, after [31]; (**b**) Geological map of the Central Kalba ore district and location of rare-metal pegmatite deposits. Numerals 1 to 3 are Gremyachy-Kina (1), Asubulak (2), and Pervomaysk-Belogorsk (3) W–E faults.

Most of the felsic dikes are composed of granite porphyry and aplite, except for the Chechek and Akhmirovo belts of ongonite dikes near Ust-Kamenogorsk city (Figure 2a), which may be facies analogs of rare-metal pegmatites. The largest Chechek dike belt encompasses approximately fifteen dikes from 2 to 5 m thick and hundreds of meters long. Muscovite phenocrysts in ongonite have an Ar-Ar age of 286 ± 3 Ma [32]. The ongonite dikes contain 1000 to 4000 ppm of rare alkalis (Li + Rb + Cs), 0.4 to 1.4 wt.% F, as well as Sn, Nb, and Ta concentration times as high as the average UCC values (15 to 100, 1.5–2, and 2–12 times, respectively). Thermobarometry reveals melts parent to ongonite with normal or high rare-metal contents. The dikes of both belts are free from greisen zones. The ongonite groundmass contains cassiterite and tantatlite-columbite, while the Sn contents are 50 to 100 times the UCC values. Thus, the Checheck and Akhmirovo ongonite dikes carry disseminated rare-metal mineralization [31].

## 3. Materials and Methods

The reported study included field and analytical work. More than 100 samples were collected from bedrock exposures from rare metal pegmatitic deposits. The mineral chemistry was analyzed on a JEOL JSM-6390LV scanning electron microscope at the D. Serikbayev East Kazakhstan State Technical University (Ust-Kamenogorsk, Kazakhstan) and on a Tescan Mira 3LMU scanning electron microscope at the V.S. Sobolev Institute of Geology and Mineralogy (Novosibirsk, Russia). The operation conditions were: 1 nA beam current, 10 nm beam diameter, and 5 μm × 5 μm spot size. The operation stability was checked against Co measurements. The detection limit was 0.1 wt.% for both major and trace elements.

The rock analyzes of Kalba granites and dikes were performed while using X-ray fluorescence analysis for major components and ICP-MS analysis for trace elements at the V.S. Sobolev Institute of Geology and Mineralogy (Novosibirsk, Russia). Complete analyzes of the rocks are published in [18,23,24,27,31].

Geochronological studies were performed by the by the $^{40}Ar/^{39}Ar$ stepwise heating method. The analyses were performed at the V.S. Sobolev Institute of Geology and Mineralogy (Novosibirsk, Russia), following the procedure that was described in [33]. The samples, which were at least 1 mm in size, wrapped in aluminum foil, vacuumed, and welded in quartz capsules, were irradiated in a Cd-lined channel of a WWR research reactor at the Tomsk Polytechnical University (Tomsk, Russia). Neutron flux was calibrated while using a reference specimen of MCA-11 biotite placed between each two samples. The neutron flux gradient did not exceed 0.5% over the sample size. Argon was released in a quartz reactor with an external heating furnace. The argon isotope composition was measured on a Micromass Noble Gas 5400 mass spectrometer.

## 4. Results

### 4.1. Rare-Metal Pegmatites: Local Geology

Almost all of the granite intrusions in the Kalba batholith coexist with more or less abundant aplitic and pegmatitic veins. Most of the pegmatite veins formed late during the emplacement of phase I and II granites of the Kalba complex. Rare-metal deposits and occurrences fall within the central and northern parts of the batholith, in a field of Li-Cs-Ta (LCT) spodumene pegmatite and ongonite dikes.

The largest pegmatitic deposits of Li, Ta, Nb, Be, and Cs are located in the Central Kalba district (Figure 2b), which includes several ore clusters (Ognevka – Bakennoe in the north, Asubulak in the center, and Belaya Gora – Baymurza in the south) along W–E faults and along the Central Kalba fault inferred from geological and geophysical data [34].

Major Ta, Nb, Be, Li, Cs, and Sn deposits are hosted by phase 1 granites of the Kalba complex crosscut by those of phase 2. Large pegmatite fields occupy margins or apophyses of intrusions and they are controlled by a system of W–E faults and pinnate subsidiary faults along them. The 400–500 m long irregularly shaped veins of pegmatites vary in thickness from 1 to 3–5 m (or even up to 20 m) and dip at 5° to 50°. Most of the large veins are highly differentiated and faulted in the axial parts.

The richest Asubulak ore cluster, with a W–E 8–10 km long and 3–4 km wide ore field, occurs in the upper part of the Tastyuba intrusion of the Kalba complex at the intersection of three faults (Figure 2b). It comprises multiple veins of aplite, aplite-pegmatite, pegmatite, and rare-metal pegmatite (Figure 3) and two W–E ore zones (Ungursai in the north and Krasny Kordon in the south) that are separated by the Asubulak fault. The deposits show successive W–E and N–S changes of mineral assemblages (Table 1).

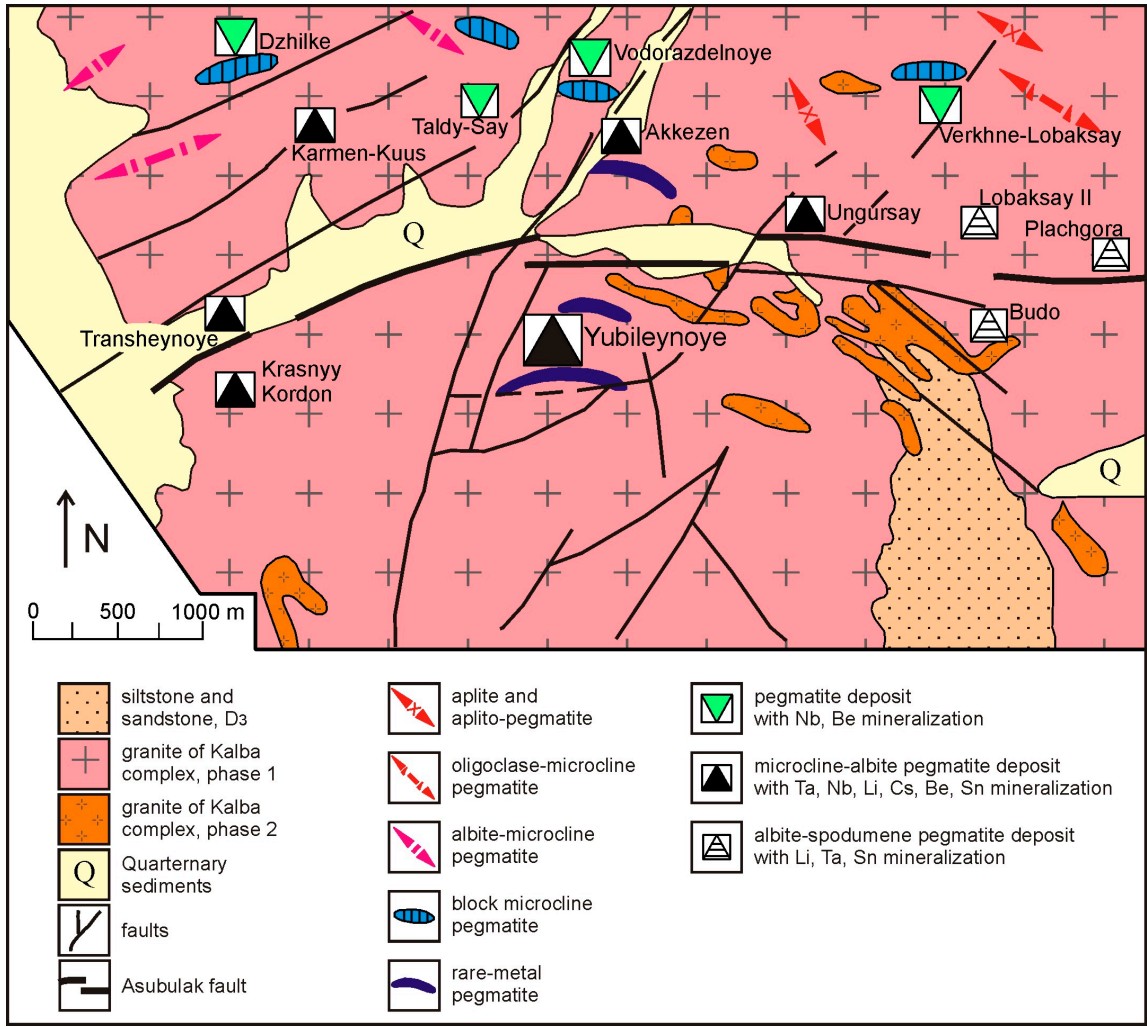

**Figure 3.** Simplified geology of the Asubulak ore cluster.

**Table 1.** Mineral assemblages in the Asubulak ore cluster: lateral zoning.

| Zone of Ore Cluster | Western Part | Center Part | Eastern Part |
|---|---|---|---|
| Pegmatite type | Microcline-albite with pollucite and petalite | Microcline-albite, albite, with spodumene | Spodumene-albite |
| Mineralization | Ta, Cs, Be, Sn | Ta, Nb, Cs, Li, Be, Sn | Li, Ta, Nb, Sn |
| Example | Karmen-Kuus, Krasny Kordon | Ungursay, Yubileynoye | Lobaksay, Plachgora, Budo |

In the N–S direction, barren oligoclase-microcline pegmatite gives way to Nb- and Be-bearing microcline pegmatite with a coarse pegmatitic texture (Taldy-Say and Vodorazdelnoye deposits) and onto rare-metal microcline-albite and albite-spodumene pegmatites (Ungursai and Yubileynoye

deposits of Ta, Nb, Be, Li, Cs, and Sn). In general, deposits in the Asubulak cluster are spaced at 1.5 km (Figure 3).

The mineralization zoning in veins reveals the following formation sequence of mineral assemblages: oligoclase-microcline pegmatite (barren) → microcline-quartz-muscovite (Nb, Be) → microcline-albite (Ta, Sn, Be) → albite (Ta, Nb, Be, Sn) → quartz-mica and albite-spodumene (Li, Ta, Be, Sn) → cleavelandite-lepidolite-pollucite-spodumene (Ta, Li, Cs, Sn).

### 4.2. Mineralogy of Pegmatites

The Kalba pegmatites show marked mineralogical and chemical diversity, with more than eighty mineral species, including rock-forming albite, quartz, microcline, and muscovite, as well as less abundant apatite, tourmaline, fluorite, and calcite. The mineral assemblages in some deposits are albite, cleavelandite, lepidolite, pollucite, spodumene, petalite, amblygonite, polychrome tourmaline, beryl, tantalite-columbite, cassiterite, etc. The main ore minerals are tantalite-columbite, cassiterite, spodumene, pollucite, and beryl.

Oligoclase-microcline assemblage: quartz, oligoclase, microcline, silver-white muscovite, prismatic black tourmaline (schorl), red or purple tetragon-trioctahedral, or less often rhombic dodecahedral garnet crystals, occasionally albite. The assemblage is predominant in barren pegmatite veins with aplite and aplite-pegmatite selvages and fine quartz-microcline-muscovite aggregates in the axes.

Microcline assemblage: fragmentary quartz-microcline blocks in large pegmatite veins, with coarse isometric crystals of microcline or their aggregates, as well as nests of coarse silver-white flaky muscovite, greenish-gray beryl, and columbite (Figure 4a); muscovite contains relatively high amounts of Ta, Nb, Be, and Sn, and rare alkalis (Li + Rb + Cs = 1.5–2%).

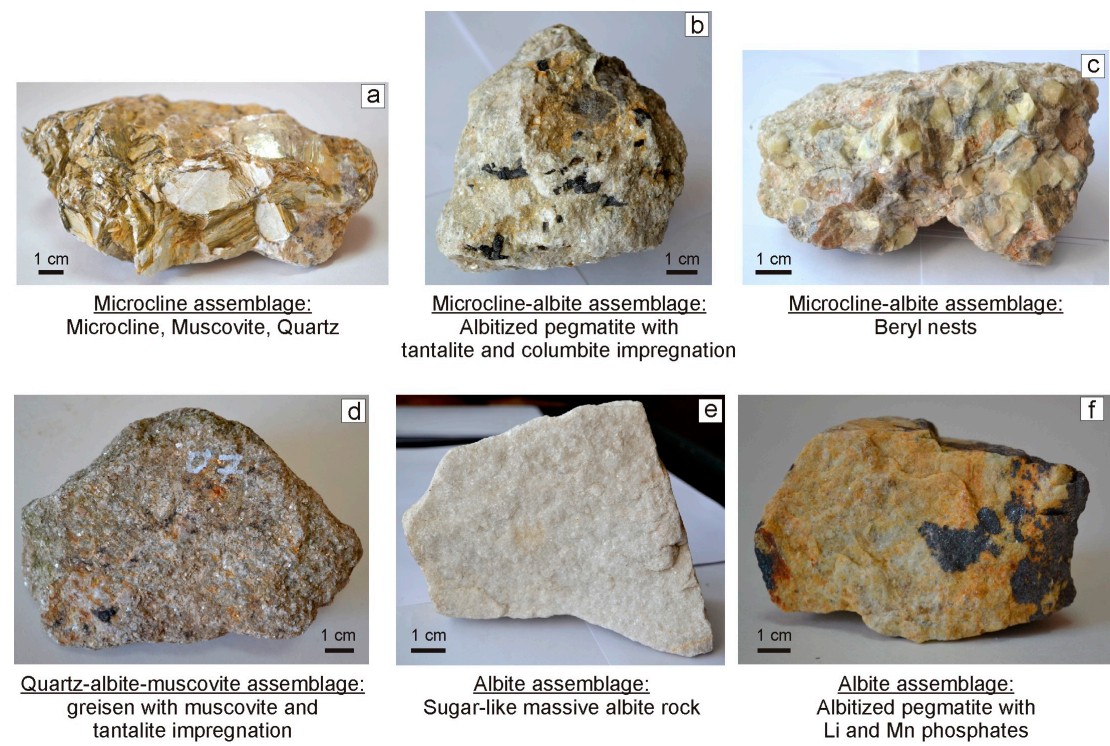

**Figure 4.** Example mineral assemblages, Asubulak ore cluster.

Microcline-albite assemblage: fine albite that replaces coarse microcline and coexists with quartz and silver-white flaky to platy muscovite, as well as with garnet and tourmaline; diverse mineralization composed of Li and Mn phosphates, sporadic spodumene crystals, radiated verdelite crystals, nests of beryl, and single crystals of tantalite and columbite (Figure 4b,c).

Albite-muscovite-lepidolite (greisen) assemblage: coarse greenish greisen (Figure 4d) of green muscovite (more than 50%), quartz, and albite, as well as fluorapatite, green tourmaline (verdelite), and sporadic spodumene; black platy crystals of tantalite-columbite that are easily extractable in ore processing; single crystals of tantalite, ixiolite $(Ta,Nb,Sn,Fe,Mn)_4O_8$, cassiterite, and other minerals. The assemblage stores the highest amount of Ta (>0.3% $Ta_2O_5$ on average) and in rare-metal rich pegmatites occurs together with the albite-spodumene assemblage.

Rare-metal albite assemblage (most widespread), carrier of Ta, Nb, Be, Li, and Sn: massive white sugar-like albite (95%) and 5% of other minerals: quartz, muscovite, garnet (Figure 4e), apatite, tourmaline (schorl and verdelite), Li-, Mn-, and Fe-phosphates (cm-scale nests and black spots, Figure 4f), columbite-tantalite (≤0.5 mm dark-brown or black prismatic crystals), and cassiterite (reddish-brown and olive-green to colorless dipyramid-prismatic crystals).

Most rare metals reside in spodumene pegmatites [28] that are composed of quartz-cleavelandite and spodumene-cleavelandite assemblages and their varieties [34]. Metal-bearing generations of minerals include cleavelandite, lepidolite, spodumene, petalite, amblygonite, pollucite, tourmaline, cassiterite, tantalite-columbite, microlite, etc., which form rich complex ore (Sn + Ta + Cs + Li) deposits.

The Supplementary Materials contains the results of the analysis of the compositions of minerals from some assemblages. Table 2 shows a comparison of the average compositions of some minerals from different assemblages. Plagioclases from the host Kalba granites (Phase 1, sample VK-1) contain from 2.4 to 5.9 wt.% CaO and up to 0.3 wt.% $K_2O$. Plagioclases from pegmatites of microcline-albite assemblage (sample VK-2) have a high-Na composition, plagioclases from rare-metal pegmatites and greisens (samples VK-5, VK-9, VK-10) are almost free of CaO and $K_2O$. Plagioclases from spodumene pegmatites (sample VK-10) also contain up to 0.37 wt.% $P_2O_5$ (see Table S1 in Supplementary Materials). K-feldspars from granites contain up to 1 wt.% $Na_2O$. K-feldspars from pegmatites contain less $Na_2O$ (about 0.3 wt.%) and more $Al_2O_3$ than K-feldspars from granites (see Table 2).

**Table 2.** Mineral composition from rock of Asubulak ore cluster.

| I. Plagioclases | | | | | | | |
|---|---|---|---|---|---|---|---|
| **Sample** | **n** | **SiO$_2$** | **Al$_2$O$_3$** | **CaO** | **Na$_2$O** | **K$_2$O** | **P$_2$O$_5$** |
| VK-1. Biotite granite. Phase 1 Kalba complex | 30 | 62.93 | 23.09 | 4.36 | 9.09 | 0.21 | – |
| VK-2. Albite-microcline pegmatite, near Yubileinoye deposit | 6 | 67.51 | 19.83 | 1.03 | 11.35 | 0.17 | – |
| VK-5. Albite pegmatite with turmaline, Yubileinoye deposit | 5 | 66.81 | 19.02 | – | 11.03 | – | – |
| VK-9. Albite-lepidolite greisen, Transheinoye deposit | 4 | 67.22 | 19.34 | 0.17 | 11.40 | – | – |
| VK-10. Albite pegmatite with spodumene, Transheinoye deposit | 9 | 67.77 | 19.74 | 0.17 | 11.67 | 0.12 | 0.33 |
| II. K-Feldspars | | | | | | | |
| **Sample** | **n** | **SiO$_2$** | **Al$_2$O$_3$** | **Na$_2$O** | | **K$_2$O** | |
| VK-1. Biotite granite. Phase 1 Kalba complex | 22 | 66.07 | 17.54 | 0.66 | | 15.20 | |
| VK-2. Albite-microcline pegmatite, near Yubileinoye deposit | 15 | 64.11 | 18.36 | 0.37 | | 16.26 | |
| VK-10. Albite pegmatite with spodumene, Transheinoye deposit | 2 | 64.54 | 18.16 | 0.33 | | 15.50 | |

**Table 2.** *Cont.*

| | | | | | | | | |
|---|---|---|---|---|---|---|---|---|
| **III. Muscovites** | | | | | | | | |
| **Sample** | **n** | **SiO$_2$** | **Al$_2$O$_3$** | **FeO** | **MnO** | **Na$_2$O** | **K$_2$O** | **F** |
| VK-1. Biotite granite. Phase 1 Kalba complex | 4 | 46.04 | 32.11 | 2.31 | – | 0.29 | 10.98 | – |
| VK-2. Albite-microcline pegmatite, near Yubileinoye deposit | 3 | 45.86 | 34.03 | 1.77 | – | 0.46 | 10.60 | – |
| VK-5. Albite pegmatite with turmaline, Yubileinoye deposit | 14 | 45.95 | 33.51 | 0.80 | – | 0.34 | 10.50 | 1.38 |
| VK-9. Albite-lepidolite greisen, Transheinoye deposit | 31 | 48.99 | 29.13 | – | 0.32 | 0.34 | 10.67 | 4.65 |
| VK-10. Albite pegmatite with spodumene, Transheinoye deposit | 26 | 45.19 | 37.07 | 0.30 | – | 0.42 | 10.57 | 0.56 |

The <u>Muscovite</u> composition is a good example illustrating the changes from granites to barren pegmatites and rare-metal rich pegmatites (see Table S3 in Supplementary Materials). Muscovite occurs sporadically in granites. Muscovites in granites (sample VK-1) contain FeO up to 3.3 wt.%, TiO$_2$ up to 1.7 wt.%, MgO up to 1.7 wt.%, and they do not contain any MnO. Muscovites in microcline-albite assemblage (sample VK-2) contains FeO up to 2.6 wt.%, TiO$_2$ up to 0.4 wt.%, MgO up to 1.0 wt.%, and only in one case 0.23 wt.% MnO. Muscovites from rare-metal pegmatites (samples VK-5, VK-10) contain less FeO, while TiO$_2$ and MgO fall below the detection limits of electron-probe analyses. In some cases, they contain 0.27 and 0.35 wt.% MnO (see Table S3 in Supplementary Materials). Muscovites from albite pegmatite with tourmaline (sample VK-5) contain up to 2.5 wt.% F and some muscovites from albite-spodumene pegmatites (sample VK-10) contain up to 1 wt.% F. Muscovites from albite-lepidolite greisen (sample VK-9) are enriched in SiO$_2$ and depleted in Al$_2$O$_3$ (see Table 2). The concentrations of FeO, TiO$_2$, and MgO are below the detection limits, while the MnO increases up to 0.44 wt.%. Muscovites from greisens are extremely enriched in F (on average, 4.65 wt.%). Some of these muscovites contain Cs$_2$O up to 1.9 wt.% (see Table S3 in Supplementary Materials).

F-apatite represents accessory apatite. Apatites from rare metal pegmatites and greisens contain appreciable amounts of Mn.

<u>Cleavelandite</u> occurs ubiquitously as curved bluish-gray platy crystals coexisting with quartz, lepidolite, apatite, spodumene, petalite, and other minerals (Figure 5a).

<u>Lepidolite</u> is a main phase of the quartz-cleavelandite-lepidolite assemblage (Figure 5b), which most often occurs in the interior of large veins. It is either (i) coarse and medium flaky coexisting with large aggregates of cleavelandite and quartz or (ii) fine to very fine flaky metasomatic quartz-lepidolite nests among other minerals. Lepidolite stores large amounts of Rb, Cs, Sn, Ta, and Nb.

<u>Spodumene</u> is among main indicator minerals of rare-metal pegmatite mineralization. It exists as thick platy yellowish-white or beige prismatic crystals (some reaching 50 cm) with vitreous luster, which become pinkish upon alteration. Some spodumene crystals form preferentially oriented textures and they coexist with quartz, cleavelandite, and amblygonite (Figure 5c,d).

<u>Amblygonite</u> (LiAl(PO$_4$)F) occurs as sporadic large elongated crystals (up to 10 cm), a typical mineral of the spodumene-cleavelandite assemblage; coexists with cleavelandite, lepidolite, spodumene, pollucite, and quartz (Figure 5e).

<u>Pollucite</u> Cs(AlSi$_2$O$_6$), which is a main phase of Cs ores, is among the latest pegmatitic minerals closely associated with Li-bearing mica, amblygonite, petalite, tourmaline, and quartz. It occurs as sugar-like milky-white fine-grained masses (Figure 5f).

<u>Tourmaline</u>, which is another indicator mineral of rare-metal ores (Ta, Nb, Sn, Li, and Cs), often forms aggregates of dark green verdellite, blue indigolite, pink rubellite (Figure 5g,h), and multicolored crystals with black heads; coexists with cleavelandite, lepidolite, and pollucite.

Tantalite-Columbite occurs as sporadic coarse black crystals (5–8 mm) in quartz, cleavelandite, lepidolite, spodumene, and other minerals, or as irregular or short-prismatic and tabular brownish-black grains (Figure 5i). Studies of the composition of tantalites-columbites from albite-lepidolite greisen (sample VK-9) showed that they contain up to 18.5 wt.% MnO (see Table S4 in Supplementary Materials).

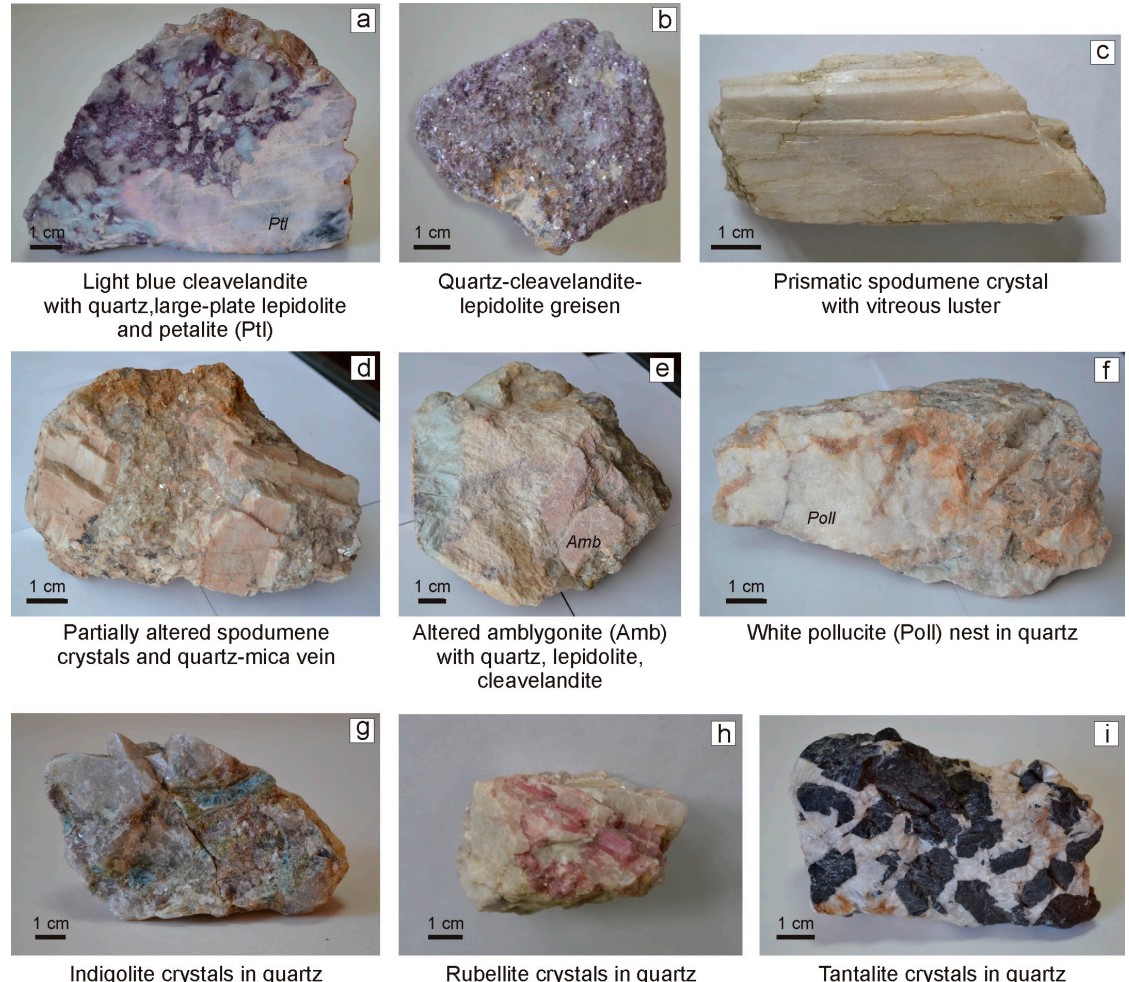

**Figure 5.** Examples of minerals from spodumene-bearing rare-metal pegmatites, Asubulak ore cluster.

Based on a review of successive mineral associations, the rare-metal mineralization of the Kalba pegmatites may be attributed to metasomatism and albitization. The latter process has produced sugar-like fine albite of new generation, medium-grained platy albite, and cleavelandite coexisting with lepidolite, pollucite, Li-tourmaline, beryl, and tantalite-columbite. The main rare-metal minerals occur on either the macro-level or as sporadic minor phases.

### 4.3. Geochronology of Rocks and Ores

According to geological data, the rare-metal pegmatites that are associated with the Kalba phase 1 granite formed around 297–296 Ma ago [18]. More rigorous age constraints have been obtained in this study with new analyses by the $^{40}$Ar/$^{39}$Ar dating on single-fraction micas present in all mineral assemblages. The obtained $^{40}$Ar/$^{39}$Ar spectra (Figure 6) show stable plateaus that correspond to 92–98% of released $^{39}$Ar.

Muscovite and lepidolite that were used for sampling were selected from different assemblages of rare-metal pegmatite. The age of greisen muscovite (Qtz-Ab-Ms assemblage) from the largest Yubileynoye deposit was 295 ± 4 Ma (Figure 6a) and lepidolite from the Cleave-Lpd-Spd assemblage was dated at 292 ± 4 Ma (Figure 6b). Two mineral assemblages from the Krasny Kordon deposit

yielded the ages of 288 ± 2 Ma for a quartz-microcline assemblage with Be mineralization (Figure 6c) and 282 ± 2 Ma for the Li-Cs-Ta-bearing cleavelandite-lepidolite-spodume assemblage (Figure 6d); lepidolite has an age of 281 ± 2 Ma (Figure 6f). The muscovite ages for two other assemblages, from the Belaya Gora deposit 10 km south of the Asubulak ore cluster are: 290 ± 2 Ma for the Qtz-Ab-Ms assemblage (Figure 6g) and 286 ± 2 Ma for that of cleavelandite-lepidolite-spodumene (Figure 6h).

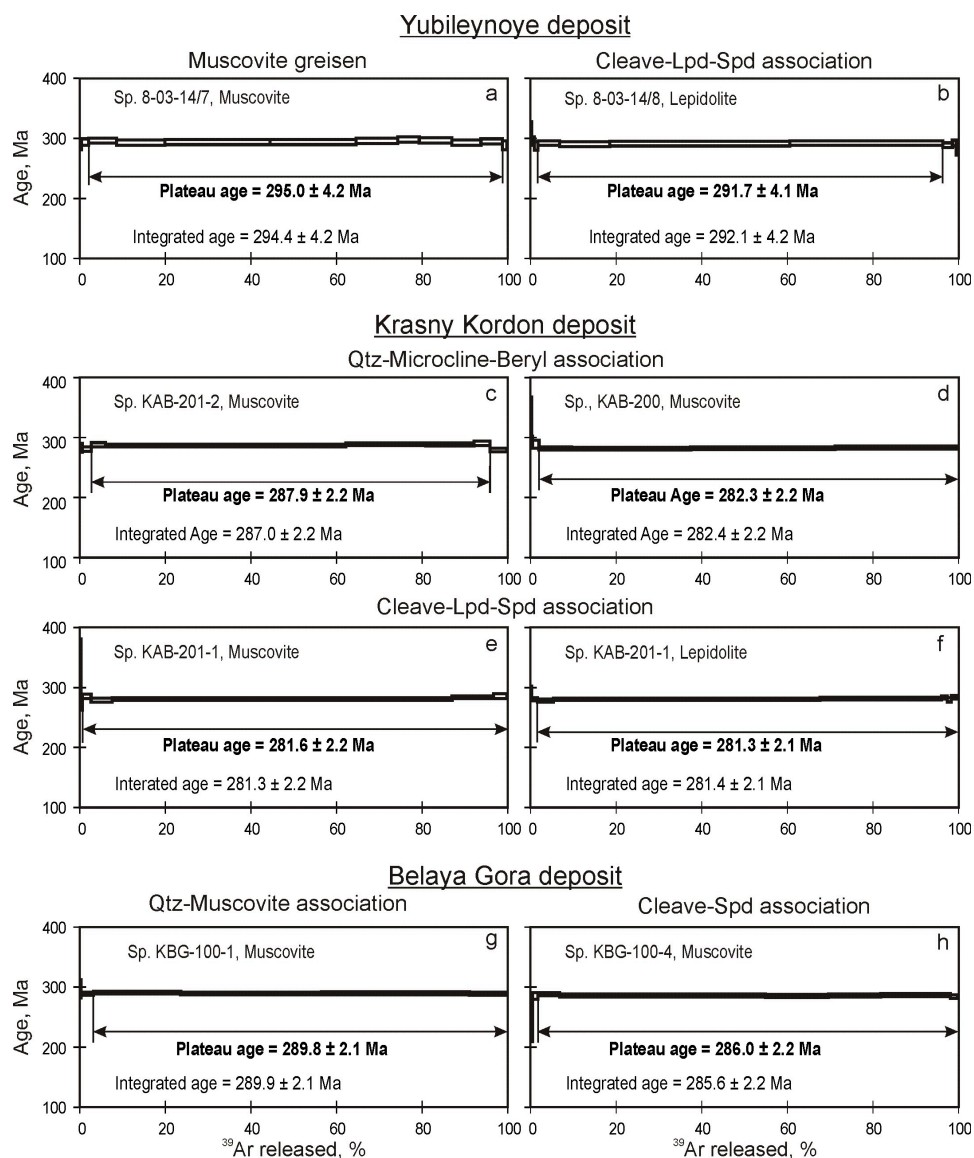

**Figure 6.** $^{40}$Ar/$^{39}$Ar age spectra of micas from rare-metal pegmatites.

Thus, the new Ar-Ar ages bracket the formation time of rare-metal pegmatites between 295 and 281 Ma.

## 5. Discussion

The geochronological data for pegmatites were compared with those for intrusions and dikes from the Kalba-Narym terrane in order to estimate the age of rare-metal mineralization in general (Figure 7). The pegmatite ages record several events of 295–292 ± 4 Ma (1), 290–286 Ma (2), and 282–281 Ma (3). The oldest dates fall within the range of the phase 1 Kalba intrusions, which contradicts the available geological evidence that the pegmatites formed at the final stage of granite emplacement or after it.

Therefore, when taking the large uncertainty in the age of the first event into account (± 4 Ma), it is reasonable to assume an age interval from 291 (295 − 4) Ma to 288 (292 − 4) Ma.

The youngest ages of pegmatite (282–281 Ma) are much younger than those of the Kalba granite and commensurate with those of the youngest Monastery granite, although one sample from the Krasny Kordon deposit showed an age of 288 Ma (Figure 6c). The K-Ar isotope system in the 282–281 Ma samples may have been upset, and a part of $^{40}$Ar was lost as a result of postmagmatic alteration or later thermal events (the closure temperature of the K-Ar system in mica was no more than 350 °C). Underestimated Ar-Ar ages were obtained for many igneous and metamorphic complexes in orogenic areas elsewhere [19,33], as well as for the Kalba granite: the age difference between the U-Pb zircon and Ar-Ar mica ages may reach 10–15 Ma [26].

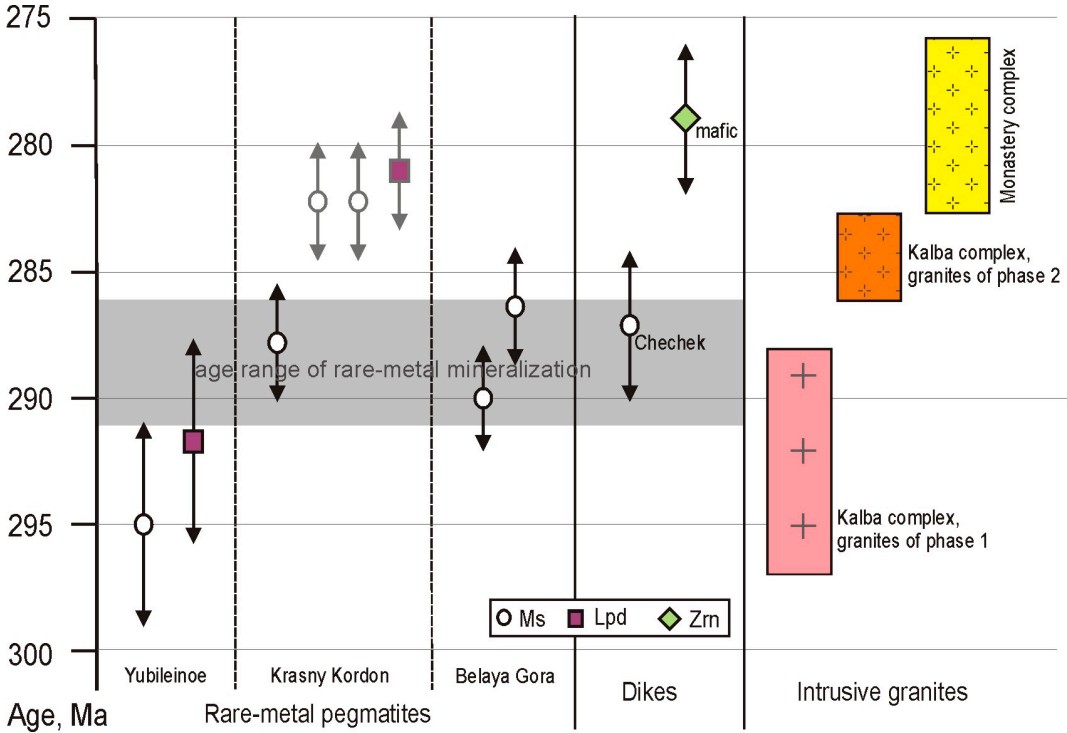

**Figure 7.** Time chart of rare-metal pegmatites, granite intrusions, and dikes in the Kalba-Narym terrane. Ages of pegmatite are as in Figure 6; ages of dikes are Ar-Ar on muscovite (Chechek intrusion), after [32] and U-Pb on zircon, after [27]; ages of granites are U-Pb on zircon for different intrusions, after [18,26,27]. Gray zone corresponds to presumable age interval of rare-metal mineralization.

More constraints on the time of rare-metal mineralization [31] are provided by the 286 ± 3 Ma $^{40}$Ar/$^{39}$Ar age of muscovite phenocrysts from a dike of rare-metal granite porphyry in the Chechek belt (Figure 2). This date can be considered to be the most accurate timing of crystallization, because dikes cool down instantaneously (on the scale of geological time), while the Chechek ongonite lack signatures of postmagmatic metasomatism [31].

Thus, the rare-metal pegmatite mineralization formed between 291 and 286 Ma (Figure 7), which is consistent with the relations of the pegmatites and the phase 1 Kalba granites that were observed in the field.

According to geological and geochronological data, the Kalba rare-metal pegmatites are related to Phase 1 granites of the Kalba complex, which may be a source of ore material (considering their relative enrichment in F, Li, Ta, and Be). The existence of granite melts with high contents of rare metals proven for ongonites of the Chechek and Akhmirovo dike belts indicates that the mineralization is of magmatic origin and it is associated with magma differentiation processes at large chambers. However, in most of intrusions in the Kalba batholith, the barren aplite, aplite-pegmatite, and quartz-feldspar

pegmatite veins occur without any mineralization that is related with granites. Rare-metal pegmatites are found under two conditions (Figures 2 and 3): at the top of large granite intrusions (i) and along large W—E and N—S faults or at their intersections (ii).

The rare-metal granite magma may have formed by two mechanisms: prolonged large-scale differentiation and inputs of mineral components. Magma differentiation was only possible in the case of voluminous phase 1 granites of the Kalba complex, because the phase 2 granites have no large pegmatite fields in their vicinity. Inputs of rare-metal mineralization could be provided by fluids that percolated through deep faults into large magma sources during the differentiation of granitic magma.

## 6. Conclusions

The reported study has provided more rigorous constraints on the local geology, the formation sequence of mineral assemblages, and ages of rare-metal pegmatite deposits in the Kalba area. The new geological and geochronological data prove that major Ta, Nb, Be, Li, Cs, and Sn deposits are associated with Phase 1 granites of the Kalba complex. The pegmatite mineral deposits formed synchronously with the emplacement of the phase 1 Kalba granites during the evolution of hydrous silicate rare-metal magmas that were produced by differentiation of granite magma at large sources with possible inputs of F and rare metals with fluids.

**Supplementary Materials:** The following are available online at http://www.mdpi.com/2075-163X/10/11/1017/s1, Table S1. Composition of plagioclases (oxides in wt. %); Table S2. Composition of K-feldspars (oxides in wt. %); Table S3. Composition of Muscovites (oxides in wt. %, formula calculated on the basis of 11 oxygen atoms); Table S4. Composition of Tantalite-Columbite (oxides in wt.%, formula calculated on the basis of 6 oxygen atoms).

**Author Contributions:** S.V.K., B.A.D., S.Z.S. and A.G.V. elaborated the subject and main idea of the paper, S.V.K., T.A.O., P.D.K., B.A.D., S.Z.S., E.N.S., O.N.K., M.A.M. and B.B.A. performed sampling, T.A.O., P.D.K., A.V.T., A.G.V., E.N.S., O.N.K., M.A.M. and B.B.A. performed sample preparation and acquired analytical data, A.V.T. performed Ar-Ar geochronological research, S.V.K., T.A.O., S.Z.S. and A.V.T. processed the acquired data, S.V.K. and T.A.O. prepared the illustrations, S.V.K., T.A.O., P.D.K., B.A.O. and S.Z.S. produced the final manuscript. All authors have read and agreed to the published version of the manuscript.

**Funding:** This research was carried out on a government assignment to the V.S. Sobolev Institute of Geology and Mineralogy (Novosibirsk) and was supported by the Russian Foundation for Basic Research (Projects no. 20-35-70076, 20-05-00346). This research was funded by the Science Committee of the Ministry of Education and Science of the Republic of Kazakhstan (Grant No. AP08856325).

**Acknowledgments:** We wish to thank Erzhan Sapargaliev for assistance in the field works. We are grateful to Alexey Kotov for his help in the minerals composition analyzing. Thanks are extended to Tatiana Perepelova for aid in manuscript preparation.

**Conflicts of Interest:** The authors declare no conflict of interest.

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
