# Peer review of "Rare-Metal Pegmatite Deposits of the Kalba Region, Eastern Kazakhstan: Age, Composition and Petrogenetic Implications"

_minerals, doi:10.3390/min10111017_

Round 1
Reviewer 1 Report
The manuscript by Sergey Khromykh and others presents a comprehensive description of the Kalba pegmatites. The Kalba region is very interesting and is an important commercial source of rare-metals. The published manuscript will be of considerable interest to mineralogists and petrologists on an international basis. But before it is published there are a few improvements that should be made.
- The pegmatites are described as the product of differentiation of granitic magma. This is the consensus view for the origin of rare-metal pegmatites such as those of the Kalba district and is based on considerable experimental evidence (Johns, Burnham, and several others, few of which are cited). However, they conclude with considerable speculation about the role of mantle-derived fluids without presenting any convincing supporting evidence. They point out that some dikes, interpreted as mantle sourced, have intersected the province but do not describe any pegmatites that have been directly intruded by such dikes or compare such pegmatites with those that haven't. They also point out that some faults have intersected the province but, again, do not compare pegmatites that have been faulted with those that haven't. Faults rarely, if ever, maintain their identity all the way down to the mantle, and typically yield to plastic deformation within the deep crust.
- In the "Materials and Methods" section they indicate "The composition of ore minerals was determined using an X-ray fluorescence ...." (line 148) but do not present quantitative mineral chemistry data or tables apart from a very few case such as muscovite (line 212). They do present a few rock analyses (lines 85, 92, 125, and 132) but do not describe how the rocks were analyzed or what standards were used.
There are also several spelling and grammar mistakes that are a bit distracting. A few examples:
Line 24 – intrusive not "intriziv"
Line 25 – insert "the" before "random"
Line 66 – insert "the" before "rocks".
Line 79 – inset "was" before "emplaced."
Line 128 – insert "the" before "felsic".
Line 323 – insert "occur" "without".
Line 324 – delete "either".
Line 345 – insert "become" before "injected" and delete "already".
Author Response
- We agree with the comments of the reviewer. We really don't have solid evidence for the mantle effect on the genesis of rare metal pegmatites. Therefore, we have corrected the conclusions. We only assume the participation of fluids enriched in F and rare metals. We exclude our first model (Figure 8) from the manuscript
- The phrase "The composition of ore minerals was determined using an X-ray fluorescence ...." was an annoying mistake. We corrected it. We also added the quantitative mineral chemistry data in two tables: the Table 2 with average mineral compositions and the Appendix A with full mineral chemistry data. We also marked in the "Materials and Methods" how the rocks were analyzed. This is a fairly large amount of data, they are published. Therefore, we decided not to provide a complete analysis of rocks in this article.
- We also have tried to correct spelling and grammar mistakes.
Reviewer 2 Report
Dear authors,
Most authors has suggested that LCT pegmatites are the result of magmatic processes from crustal-derived melts. You cannot demostrate that the origin of the mineralization of typical LCT pegmatites is produced from "mantle-derived fluids" with the facts and data you have presented.
I understand that "other authors" in "other places", this hipothesys has been sugested by using isotopic Sr and Nd compositions. Thus, if you want to suggests the same explanation, you have to use the same demonstration. It is not enough the coexistance with mafic dikes, in which this type mineralizationis absent.
It is totally useless to discusse about some ideas that you cannot demonstrate.
Best regards,
Author Response
We agree with the comments of the reviewer. We really don't have solid evidence for the mantle effect on the genesis of rare metal pegmatites. Therefore, we have corrected the conclusions. We only assume the participation of fluids enriched in F and rare metals. We exclude our first model (Figure 8) from the manuscript
Round 2
Reviewer 2 Report
Dear authors, I think now your work is much better now, and also it is consitent with most of the published research on granitic pegmatites. But if you cannot demostrate the participation of mantle-derived fluids in P-F-rich pegmatites by isotopic signatures, the lines from 366 to 373 should be deleted. Best regards.
Author Response
Dear Reviewer!
We agree that we cannot cite isotope data and demonstrate conclusive evidence for mantle-derived fluids in pegmatites. Ok, we've removed our reasoning about this by clearing lines 366-373. References were also shortened accordingly.